# Cardiovascular and Metabolic Protection by Vitamin E: A Matter of Treatment Strategy?

**DOI:** 10.3390/antiox9100935

**Published:** 2020-09-29

**Authors:** Melanie Ziegler, Maria Wallert, Stefan Lorkowski, Karlheinz Peter

**Affiliations:** 1Department of Cardiology and Angiology, Internal Medicine III, University Clinic of Tübingen, 72076 Tübingen, Germany; melanie.ziegler@med.uni-tuebingen.de; 2Institute of Nutritional Sciences, Friedrich Schiller University, 07743 Jena, Germany; maria.wallert@uni-jena.de (M.W.); stefan.lorkowski@uni-jena.de (S.L.); 3Competence Cluster for Nutrition and Cardiovascular Health (nutriCARD) Halle-Jena-Leipzig, Germany; 4Atherothrombosis and Vascular Biology Laboratory, Baker Heart and Diabetes Institute, 75 Commercial Road, Melbourne, VIC 3004, Australia; 5Department of Medicine and Immunology, Monash University, Melbourne, VIC 3800, Australia; 6Department of Cardiometabolic Health, University of Melbourne, Melbourne, VIC 3800, Australia; 7Department of Cardiology, The Alfred Hospital, Melbourne, VIC 3800, Australia

**Keywords:** vitamin E, cardiovascular disease, myocardial infarction, risk factors, treatment strategy

## Abstract

Cardiovascular diseases (CVD) cause about 1/3 of global deaths. Therefore, new strategies for the prevention and treatment of cardiovascular events are highly sought-after. Vitamin E is known for significant antioxidative and anti-inflammatory properties, and has been studied in the prevention of CVD, supported by findings that vitamin E deficiency is associated with increased risk of cardiovascular events. However, randomized controlled trials in humans reveal conflicting and ultimately disappointing results regarding the reduction of cardiovascular events with vitamin E supplementation. As we discuss in detail, this outcome is strongly affected by study design, cohort selection, co-morbidities, genetic variations, age, and gender. For effective chronic primary and secondary prevention by vitamin E, oxidative and inflammatory status might not have been sufficiently antagonized. In contrast, acute administration of vitamin E may be more translatable into positive clinical outcomes. In patients with myocardial infarction (MI), which is associated with severe oxidative and inflammatory reactions, decreased plasma levels of vitamin E have been found. The offsetting of this acute vitamin E deficiency via short-term treatment in MI has shown promising results, and, thus, acute medication, rather than chronic supplementation, with vitamin E might revitalize vitamin E therapy and even provide positive clinical outcomes.

## 1. Introduction

Cardiovascular diseases (CVD) such as atherosclerosis are a major cause of mortality and morbidity worldwide. Multiple factors are involved in the complex etiology of atherosclerosis. One of the most important factors that drives atherosclerosis and its complications, such as myocardial infarction (MI) and stroke, is inflammation. Inflammation plays a pivotal role in both the initial, as well as the advanced, phases of atherosclerosis, including plaque destabilization and ultimate rupture. In particular, the advanced phase of atherosclerosis is characterized by a high degree of inflammation that includes high-level production of reactive oxygen species (ROS), due to excessive oxidative stress [1,2]. One of the major initial inducers of oxidative stress in atherosclerosis is pathological shear stress levels and flow patterns [3]. An imbalance between oxidants and antioxidants enhances oxidative stress and promotes oxidation of circulating low-density lipoprotein cholesterol (LDL-C). Elevated oxidized LDL-C (oxLDL) acts as a damage-associated molecular pattern and damages the endothelium further, and is also internalized by monocytes/macrophages, which consequently transform into foam cells and amplify the inflammatory process further [2]. Consequently, the secretion of pro-inflammatory chemokines and cytokines, and the activation of immune cells, exacerbate the progression of atherosclerosis. Antioxidants, either endogenous antioxidants like superoxide dismutase (SOD), catalase, glutathione (GSH) and GSH peroxidase, or antioxidants derived from dietary sources like vitamin A, C, and E, are needed to scavenge free radicals and other toxic radicals [4]. Patients suffering from inflammatory diseases often present with decreased levels of antioxidants, due to either insufficient dietary intake or increased demand for antioxidants [5]. Vitamin E is a very potent antioxidant substance, and shows high anti-inflammatory properties [6,7]. Therefore, vitamin E, particularly the α-tocopherol (α-TOH) form, has been suggested as a promising candidate in the prevention of CVD. However, enthusiastic research on vitamin E in large clinical trials has only resulted in controversial and mostly discouraging outcomes, and ultimately has not provided evidence for overall beneficial effects of vitamin E in CVD, with a few exceptions, as discussed below. The aim of the present review is to critically summarize the data available on vitamin E supplementation in diseases in general and systematically investigate potential reasons for the observed conflicting results, and we also provide a perspective on what we have learned from the past trials for future trials. We ultimately redirect the focus from chronic vitamin E supplementation to short-term vitamin E medication in acute clinical settings caused by high inflammatory and oxidative stress, such as MI.

## 2. Vitamin E Characterization

Vitamins are micronutrients essential for the maintenance of metabolic functions, but not necessarily for energy regeneration. Humans are not able to synthesize vitamins by themselves, so vitamins need to be supplied from food. The function of vitamin E as a vitamin was first discovered in 1922 by Evans and Bishop, demonstrating its relevance as a fertility factor in rats [8]. Later the α-TOH form, more precisely the stereoisomere *RRR*-α-TOH, of vitamin E was identified to convey this property. The genetic disorder ataxia with vitamin E deficiency results in low levels of α-TOH, due to a defect in the α-tocopherol transport protein (α-TTP). Only the *RRR*-α-TOH form serves as a proper treatment for this disease. Therefore, there is an ongoing discussion whether only *RRR*-α-TOH conveys the vitamin function and other forms of vitamin E would not [9].

### 2.1. Different Forms of Vitamin E

The group of vitamin E consists of α-, β-, γ-, and δ-forms of tocopherols (TOH) and tocotrienols (T3) [9] (Figure 1). The course of vitamin E digestion follows in general the course of coexisting lipid compounds. Pancreatic and intestinal enzymatic digestion followed by circulation and distribution to the liver and non-hepatic tissues is the same for all vitamin E forms. To date, no discrimination of α-TOH over non-α-TOH forms is known for intestinal absorption [10], whereas α-TTP in the liver discriminates in favor of α-TOH. Consequently, α-TOH is the most abundant form of vitamin E in healthy humans [11]. In humans, circulating α-TOH concentrations of >12 mmol/L are defined as adequate, whereas <12 mmol/L represents a marginal and <9 mmol/L a severe deficiency [12,13]. Today, biological activity is best investigated for α-TOH. However, the biological relevance of non-α-TOH forms of vitamin E is increasingly becoming the focus of research, due to observed, at least in part, distinct properties compared to α-TOH.

### 2.2. Metabolism of Vitamin E

Hepatic degradation of all vitamin E forms is inversely regulated to the amount of vitamin E supplied in the diet, thus, protecting from excessive accumulation of vitamin E [14]. α-TTP preferentially binds to α-TOH in contrast to non-α-TOH forms, which protects α-TOH from excessive degradation in the liver and excretion via feces and urine [15,16]. α-TOH is mainly released into the circulation by the liver in nascent lipoproteins, such as very-low-density lipoprotein (VLDL), whereas non-α-TOH forms are primarily metabolically degraded followed by their excretion [10,17,18]. Side-chain truncation of vitamin E initiates the metabolism by cytochrome P450 (CYP) 4F2/3A4-dependent ω-hydroxylation, resulting in the formation of long-chain metabolites (LCM) of α-TOH, α-13′-OH [19,20] and α-13′-COOH. The catabolic end-product of vitamin E degradation is α-carboxyethyl hydrochroman (α-CEHC) [21]. Excessive supplementation of α-TOH increases α-TOH plasma levels and enhances degradation of both non-α-TOH forms and α-TOH itself, and subsequently the excretion of the respective CEHC [22]. Hence, α-CEHC is used as a marker for α-TOH status in healthy humans [23,24]. Although catabolism is more rapid for T3s than TOHs and varies within the TOHs, the metabolic pathway is the same [21].

### 2.3. Relevance and Evidence of Vitamin E as an Antioxidant/Pro-Oxidant and Its Non-Antioxidative Properties

The vitamin E molecules consist of three functionally distinct domains: the functional domain, the signaling domain, and the hydrophobic domain [25], mediating the antioxidative capacity and non-antioxidative effects of vitamin E. All vitamin E forms have the hydroxyl group at position C6 in common, which results in similar antioxidative capacity for all vitamin E forms [26]. Vitamin E, more precisely α-TOH, protects or promotes oxidation depending on prevailing oxidative conditions. Under strong oxidative conditions, α-TOH is known as the most important lipid-soluble radical chain–breaking antioxidant, protecting phospholipids in cell membranes and plasma lipoproteins against peroxidation of polyunsaturated fatty acids (PUFA), at least in vitro [7,27]. Under mild oxidative conditions, the tocopheroxyl radical is regenerated to intact tocopherol by co-antioxidants [28]. The most important co-antioxidant is ascorbic acid, which subsequently forms an ascorbyl radical under the regeneration of α-TOH. With a deficiency of co-antioxidants [28] or high concentrations of α-TOH, α-TOH has been discussed to promote pro-oxidant properties. Even in vivo vitamin E supplementation was described as a potential risk factor for increasing gastrointestinal symptoms and total mortality [29], possibly induced by increased oxidative activity in plasma [30]. However, a study by Miller et al. reporting increased mortality caused by α-TOH supplementation was controversially discussed [31]. Subsequent evaluation using different methodological approaches to meta-analysis did not show a causal relationship between vitamin E supplementation and increased mortality [31]. In addition, in studies in healthy subjects not showing increased oxidative stress, supplementation of α-TOH did not affect markers of oxidative damage [32], while high-dose supplementation of α-TOH even decreased oxidative stress in hypercholesterolemic patients [33,34]. So far, no clinical trial has observed adverse effects of vitamin E supplementation in healthy subjects [35].

In the early 1950s, Hickman and Harris first mentioned properties of vitamin E independent from its antioxidative capacity [36]. Decades later, the research of Azzi and colleagues established several working hypotheses regarding the non-antioxidative properties of α-TOH [37]. To date, α-TOH and non-α-TOH forms are known to regulate the expression of genes and proteins, the activity of enzymes, signaling cascades within uptake, transport, degradation, metabolism and excretion of vitamin E forms, lipoprotein uptake, and inflammation, to name only a few of their functions [27]. These non-antioxidative properties control several events in atherosclerosis, such as the inhibition of smooth muscle cell (SMC) proliferation, the preservation of endothelial integrity, the inhibition of monocyte–endothelial adhesion, the inhibition of monocyte ROS and cytokine release, and the inhibition of platelet adhesion and aggregation [37]. In addition, scavenger receptors, adhesion molecules, and collagenase seem to be under the non-antioxidant control of α-TOH [38].

## 3. Cardiovascular Diseases

### 3.1. Atherogenesis, Atheroprogression, and Plaque Rupture

The earliest event in atherogenesis is endothelial dysfunction triggered by a number of insults including disturbed flow patterns and low shear stress, arterial flow, hypertension, homocysteinemia, hyperlipidemia, diabetes, physical injury, and ROS [39]. Endothelial dysfunction leads to a disrupted endothelial barrier, contributing to lipid accumulation within the intima and so-called fatty streak formation. The inflamed endothelium expresses adhesion molecules and secretes chemokines, leading to leukocyte recruitment. At this point, monocytes transmigrate through the endothelium into the intima, differentiate into macrophages, absorb the lipoproteins, and become foam cells. The last key step is the degradation of the extracellular matrix, and then an atherosclerotic plaque is formed [40]. A stable atherosclerotic plaque has a lipid-rich necrotic core encapsulated by a thick and stable fibrous cap. As the plaque progresses, the core becomes more necrotic and larger in size, and more intraplaque hemorrhage occurs, as well as thinning of the fibrous cap; this advanced plaque is called an unstable or vulnerable plaque [41]. At this stage, the plaque is very fragile with a high risk of rupture. Plaque rupture exposes thrombogenic components, leading to thrombosis, which can then cause occlusion of the respective or a downstream artery resulting in an acute MI or stroke [42,43,44].

### 3.2. Cardiovascular Events

CVD are the leading cause of mortality worldwide, representing 31% of all global deaths [45]. CVD include coronary artery disease, cerebrovascular disease, peripheral arterial disease, deep vein thrombosis, and pulmonary embolism. Most cardiovascular events and deaths can be explained by risk factors such as hypertension, dyslipidemia, diabetes, obesity, unhealthy diet, physical inactivity, and smoking. Strikingly, epidemiological studies have shown that 75% of premature CVD are preventable by early intervention [46,47]. Coronary heart disease (CHD), also called ischemic heart disease (IHD) and coronary artery disease, is the CVD with the highest morbidity and mortality worldwide [45,47]. In CHD, the blood supply is mostly reduced or blocked due to atherosclerotic plaques and its complications, particularly an acute atherothrombotic event in the coronary arteries. The dynamic nature of CHD results in various clinical manifestations which can be categorized as either acute coronary syndrome, such as ST-elevation MI (STEMI) and non-ST-elevation MI (NSTEMI), or chronic coronary syndrome, such as stable angina pectoris [48].

Myocardial infarction is the single leading cause of death globally [45]. It is typically defined by chest pain, abnormal cardiac biomarkers (typically troponin I or T), and potentially typical electrocardiogram (ECG) changes (ST-elevation) [49]. The treatment of MI with timely percutaneous coronary intervention (PCI) or pharmacological thrombolysis has contributed to substantial improvement in the outcome of patients suffering from MI. However, paradoxically, restoration of coronary blood flow causes further myocardial damage, referred to as ischemia/reperfusion injury [50]. The early phase of ischemia/reperfusion injury is characterized by extensive tissue damage to the myocardium caused by inflammatory and oxidative stress responses leading to immune cell infiltration, as well as production and release of pro-inflammatory cytokines and chemokines. Imaging the area of oxidative stress defines the ischemic and reperfused myocardium very well [51]. Following this, the late phase consists of wound healing and fibrotic repair [52].

## 4. Vitamin E and Risk Factors for Cardiovascular Events

The association between vitamin E and risk factors for cardiovascular events will be discussed in detail in the following chapter, and is summarized in Figure 2 and Table 1.

### 4.1. Hypertension

Hypertension, defined as a systolic blood pressure (SBP) ≥ 130 mmHg and/or a diastolic blood pressure (DBP) ≥ 80 mmHg, is a major public health burden worldwide [98] and the most prevalent cause of CVD, but only about 50% of patients are adequately treated and achieve adequate blood pressure (BP) control.

An observational study by Kuwabara et al. [53] re-analyzed the data from the National Health and Nutrition Survey 2007 and studied the relationship between hypertension and dietary intake of vitamin E. In this study, data from 2102 females and 1405 males were analyzed. Higher vitamin E intake was associated with lower prevalence of hypertension. A recent meta-analysis including 839 participants in 18 clinical trials suggested that vitamin E supplements significantly decreased only SBP (15 clinical trials). However, vitamin E supplementation had no favorable effect on DBP (12 clinical trials) or mean arterial pressure (five clinical trials) [99]. Within this meta-analysis, the characteristics of the patients were very diverse. Vitamin E doses varied from 134 to 1206 mg per day and the duration of supplementation varied between 3 and 48 weeks. Focusing on the hypertensive trials only, Boshtam et al. showed that vitamin E reduced blood pressure in mild hypertensive participants [54]. In a randomized, triple-blinded study, 70 mild hypertensive patients were allocated into a vitamin E supplement (134 mg per day (200 IU per day)) or a placebo group. After 27 weeks, vitamin E supplementation caused a remarkable decrease in SBP (−24% versus −1.6% in placebo) and a less remarkable decrease in DBP (−12.5% versus −6.2% in placebo). A more recent study also demonstrated a decrease in blood pressure after 12 weeks of vitamin E supplementation (134 mg per day) in mild hypertensive subjects [55]. Palumbo et al. demonstrated in 142 treated hypertensive patients that 12 weeks of vitamin E supplementation (300 mg per day) showed no clinically relevant effect on blood pressure [56]. A further study by Mihalj et al. [57] also showed no further effect of vitamin E supplementation in patients treated with antihypertensive therapy, in this case an angiotensin II receptor blocker. Barbagallo et al. showed no effect of vitamin E treatment on SBP or DBP in 12 hypertensive patients treated with 600 mg vitamin E per day for 4 weeks in comparison to a placebo control group [58]. The effect of vitamin E supplementation (500 mg per day for 6 weeks) on high blood pressure variability was studied by Hodgson et al., and they showed no significant alteration in the rate of blood pressure variation [100]. Often, vitamin E supplementation is studied in combination with vitamin C. Mellyana et al. [101] demonstrated that a combination of vitamin E and C supplementation improved blood pressure in pediatric idiopathic nephrotic syndrome patients.

Despite promising preclinical models and observational studies, antioxidant strategies for the treatment of hypertension in the clinic has not achieved the expected success [102]. The use of vitamin E and other antioxidants in the treatment of hypertension, including the antioxidant effects of antihypertensive drugs, is controversial [103,104]. More convincing effects were shown in participants with mild hypertension, while trials with treated hypertension or uncontrolled hypertension have not shown benefits of vitamin E supplementation. One reason for this could be the complexity of this condition, as hypertensive patients are remarkably diverse, ranging from the young and lean to the obese and elderly. In each of these phenotypes, the role of vitamin E, as well as other oxidants, might be different and therefore diluted within clinical trials that do not select appropriate patient groups [102]. Furthermore, the duration and the dosage of supplementation of α-TOH vary and might be important factors in successful therapy. Finally, oxidative stress might not be the cause but, rather, the consequence of hypertension in humans [57].

In conclusion, promising preclinical and observational studies of vitamin E supplementation in hypertensive disease have not been broadly replicated in randomized controlled trials (RCT, Figure 2). However, in the subgroup of patients with mild hypertension, vitamin E supplementation seems to provide some benefits.

### 4.2. Hyperlipidemia

Currently, the most central approach to reducing the number of cardiovascular events is the lowering of circulating LDL-C, based on clear evidence of LDL-C being a strong causal risk factor for CVD [105,106]. Statins are the backbone of the therapy for hyperlipidemic patients [107]. They inhibit the β-hydroxy-β-methylglutaryl-coenzyme-A (HMG-CoA) synthetase, with atorvastatin, simvastatin, pravastatin, and lovastatin being the most common statins in use. Statins are categorized by their LDL-C-lowering capacity, and high-, moderate-, and low-intensity statins lower LDL-C by ≥50%, 30–49%, and <30%, respectively [107]. There are controversial findings on statins lowering vitamin E concentration [108]. Whereas some studies showed a reduction due to the LDL-C-lowering effect of statins, others did not. In addition, the pleiotropic effects of statins show anti-inflammatory, antioxidative, and anti-thrombotic properties. Hypercholesterolemic patients with increased oxidative stress showed lower vitamin E plasma levels compared to healthy subjects, potentially counterbalancing the increased oxidative stress level [59]. As reported by Cangemi et al., treatment with atorvastatin (10 mg/d), for 3 d and 30 d, in hypercholesterolemic patients, respectively, decreased oxidative stress, measured by urinary isoprostanes, and restored levels of vitamin E/cholesterol [59]. These findings were supported by Lui et al.’s study showing atorvastatin (10 mg/d for 5 months) increased circulating vitamin E/LDL-C in hypercholesterolemic patients; however, the urinary oxidative stress marker 8-OHdG remained unchanged under atorvastatin treatment [62]. Simvastatin (20–40 mg/d) administered for 8 weeks in hypercholesterolemic patients significantly increased the levels of lipid-corrected α-TOH and γ-TOH [60]. These findings support the hypothesis that statins possess antioxidative properties so that fewer additional antioxidants are needed. Therefore, additional supplementation with antioxidants in statin-treated patients will most likely not result in further decrease in oxidative stress [109,110].

Furthermore, clinical findings suggested that there might be an interaction between statins and vitamin E, since the regression of coronary artery disease observed with a combined therapy of simvastatin and niacin was lost in combination with vitamin E [63]. Thus, the effects of vitamin E on statin bioavailability and decreased therapeutic efficiency were suggested, since vitamin E and statins share similar hepatic metabolic enzymes. In addition to CYP4F2, vitamin E is metabolized by CYP3A4, the same enzyme system used for hepatic xenobiotic metabolism of simvastatin and lovastatin. However, Leonard et al. reported that vitamin E supplementation did not alter cholesterol levels under statin therapy [63].

Other therapeutic approaches to lowering LDL-C include the drug ezetimibe and the lowering of the proprotein convertase subtilisin/kexin type 9 (PCSK9) by either antibodies or genetic means. PCSK9-targeted therapy lowers LDL-C to concentrations of <15 mg/dL by targeting LDL-C receptors for lysosomal degradation, instead of allowing them to be recycled to the hepatocyte cell surface. Initially with the anti-PCSK9 antibody, there was concern that this aggressive LDL-C lowering might lead to steroid hormone and vitamin E deficiency. Vitamin E is transported in the circulation via LDL-C particles. Consequently, reduction of LDL-C was expected to decrease vitamin E levels in circulation. However, this assumption was not confirmed [111]. Vitamin E concentration in red blood cells and in plasma remained unaltered [61], although LDL-C decreased to <40 mg/dL for 87% and even to <15 mg/dL for 40% of patients treated with evolocumab, a clinically used anti-PCSK9 antibody [61,111].

Consequently, the outcomes of large clinical trials, indicating that there is no beneficial effect of vitamin E supplementation, need to be analyzed with caution, as concomitant use of statins may have reduced vitamin E’s effectiveness [112].

In summary, lipid-lowering therapies do not decrease circulating vitamin E, but the opposite. In addition to lipid-lowering, statins exert anti-inflammatory and antioxidative effects. Consequently, decreased vitamin E plasma levels in hypercholesterolemic patients with increased oxidative stress can be restored by statins. However, additional beneficial effects on the CVD risk of supplemented vitamin E in patients receiving lipid-lowering therapies have not been observed. Metabolic interactions of vitamin E and statins that affect statins’ bioavailability cannot be excluded, and are of special importance in the therapeutic efficiency of statins (Figure 2).

### 4.3. Thrombosis

Thrombosis is the formation of a blood clot within blood vessels, both venous and arterial, leading to a partial or complete blockage of the vessel. Here, we mainly discuss studies focusing on vitamin E and venous thrombosis, as arterial thrombosis is discussed in Section 3.2. Regular doses of vitamin E may reduce the risk of venous thromboembolism in women. Glynn et al. [64] reviewed data from 39,876 women aged 45 and older taking part in the Women’s Health Study. Women were randomly allocated to take either a regular dose of vitamin E (600 IU) or a placebo on alternate days over a 10-year period. The data indicated that, in general, women taking vitamin E were 21% less likely to suffer a venous thromboembolism. The study authors cautioned that more research is needed to confirm this link in the prevention of venous thromboembolism.

A large case-controlled study by Vuckovic and colleagues [65] included 2506 patients with venous thrombosis, 2506 partner controls, and 2684 random-digit-dialing (RDD) controls. Only 96 patients were supplemented with vitamin E, while the other patients were supplemented with other vitamins or a multivitamin supplement. This showed that after extensive adjustments, vitamin E supplementation was no longer associated with a decreased venous thrombosis risk. However, this study lacks information about the duration and dose of vitamin E supplementation, so it is unknown whether vitamin E supplementation would decrease venous thrombotic risk in people with low vitamin E baseline levels.

Furthermore, the interaction of vitamin E and K may affect thrombosis. The metabolic pathways of vitamin E and K showed parallels e.g., side chain ω-hydroxylation by Cytochrom P450 4F2 (CYP4F2) and the following β-oxidation. Further pregnane X receptor (PXR) binds vitamin E and vitamin K and activates genes involved in xenobiotic detoxification such as multiple drug resistance 1 (MDR1) and CYP3A4 [113,114,115,116,117]. In addition, PXR binds the vitamin K metabolite menaquinone-4. The PXR/menaquinone-4 complex regulates collagen formation and extracellular matrix in osteoblastic cells [118]. The active form of vitamin K, as an important regulator for blood coagulation, is altered by excessive vitamin E. Hence, vitamin E has been suggested for therapeutic use in patients with increased risk of thrombosis [113].

Furthermore, the interaction of vitamin E and K may affect thrombosis. The metabolic pathways of vitamin E and K overlap with each other, e.g., side chain ω-hydroxylation by Cytochrom P450 4F2 (CYP4F2) and the consequent β-oxidation. Furthermore, pregnane X receptor (PXR, also known as SXR) binds vitamin E and vitamin K and activates genes involved in xenobiotic detoxification such as multiple drug resistance 1 (MDR1) and CYP3A4 [113,114,115,116,117]. In addition, PXR binds the vitamin K metabolite menaquinone-4. The PXR/menaquinone-4 complex regulates collagen formation and extracellular matrix in osteoblastic cells [118]. Therefore, vitamin K2 may serve as a critical factor regulating bone matrix formation. Vitamin E may bind to the same receptor, however, so far there is no indication that high-dose vitamin E supplementation causes adverse effects on bone homeostasis [119]. The active form of vitamin K, as an important regulator for blood coagulation, is altered by excessive vitamin E. High-dose vitamin E supplementation increased PIVKA-II in adults not receiving oral anticoagulant therapy. An increase in proteins induced by vitamin K absence–factor II (PIVKA-II) is an indicator of a poor vitamin K status. In contrast, other measures of the vitamin K status e.g., plasma phylloquinone concentration and percentage of ucOC did not change significantly in response to the vitamin E supplementation [120]. The finding that the higher the vitamin E serum levels, the higher the risk of bleeding has been related to competitive metabolism of vitamin E and K, and high doses of vitamin E may antagonize vitamin K [121]. Vitamin E has even been discussed for therapeutic use in patients with increased risk of thrombosis [113].

In summary, the number of studies focusing on the effect of vitamin E supplementation on thromboembolic risk is low, and to our knowledge there is no study directly reporting on vitamin E plasma levels and the associated risk of venous thromboembolism. As such, further studies are needed to answer the question of a potential risk reduction in venous thromboembolisms by vitamin E supplementation (Figure 2).

### 4.4. Chronic Inflammation

#### 4.4.1. Age

With increasing age, the burden of elevated oxidative stress promotes both infections and non-communicable diseases such as CVD, cancer, Alzheimer’s disease (AD), and diabetes mellitus type 2 (DMT2). In short, aging is associated with chronic low-grade inflammation, which alters the health status of elderly subjects. Partly inadequate nutritional intake of macro- and micronutrients by the elderly confounds this finding. Elderly people living independently must also be distinguished from subjects living in aged-care or similar institutions [122]. Critical nutritional deficits were observed concerning antioxidant micronutrients, thus forcing an imbalance between enhanced oxidative processes during aging and levels of antioxidants. As reported in several studies, the plasma concentration of α-TOH is decreased in older adults, associated with a higher incidence of infections such as upper respiratory infections [123], non-communicable diseases, and impairment of cognitive function [66]. Other studies have observed an inverse correlation of α-TOH plasma level in patients with DMT2, but not in age-matched healthy subjects [67,124]. In contrast, the plasma concentration of γ-TOH and concentrations of α-TOH and γ-TOH in platelets, as well as total tocopherol concentrations, decreased significantly with age [67]. However, the hypothesis cannot be completely excluded that lower nutritional intake of α-TOH causes the observed deficiencies, rather than increased consumption of α-TOH due to higher oxidative stress [68], although age-related differences in vitamin E metabolism were also suggested [125]. The majority of institutionalized elderly subjects have been reported to consume less than two-thirds of the recommended daily intake (RDI) of α-TOH [69,70]. The current RDI level of vitamin E is 15 mg/d for subjects >14 years, regardless of varying circumstances during aging. The German Society of Nutrition recommends 11 mg/d for women and 12 mg/d for men >65 years. Therefore, Meydani and colleagues recently raised the question of whether the RDI of vitamin E should be adjusted, thus improving immune and inflammatory responses, and reducing the risk of age-related diseases [126,127], such as poor mental health [68]. A daily intake level of 200 IU α-TOH has proven to increase immunity, more precisely T-cell function, in elderly individuals [71,127].

In summary, several studies have reported critical antioxidant deficits in the elderly. As potential causes, a lower nutritional intake of α-TOH, mostly observed in institutionalized elderly subjects, and increased consumption of α-TOH, due to higher oxidative stress with increasing age, have been discussed. Comorbidities often contribute to a-TOH deficiency (Figure 2).

#### 4.4.2. Obesity

Obesity is a global pandemic, with increasing incidences being most alarming in children and adolescents. Data from the US National Health and Nutrition Examination Survey (NHANES) showed that one-third of adults and 12.5 million children and adolescents in the United States are classified as obese. Over 1999–2018 prevalence of overall obesity increased from 30.5% to 42.4% and severe obesity increased from 4.7% to 9.2% [128]. The consequences of this are dire, since obesity is an independent, strong risk factor for DMT2 and CVD. Obesity is accompanied by low-grade inflammation triggered by increased oxidative stress and lipid or protein oxidation [129,130]. Therefore, there is a clear need for an adequate supply of antioxidants, such as vitamin E. Studies reporting on the nutritional intake of vitamin E have found no differences in vitamin E intake in obese compared to non-obese subjects [75,129] or increased α-TOH intake due to the nature of vitamin E as a fat-soluble vitamin [72]. Intake of monounsaturated fatty acids (MUFA) and PUFA is increased in obese subjects, resulting in increased total cholesterol (TC), LDL-C and triglycerides (TG), and vitamin E intake, whereas high-density lipoprotein cholesterol (HDL-C) is decreased [129,130]. However, intake of vitamin E does not correlate with vitamin E plasma level [72]. Lipids, more precisely PUFA and MUFA, are prone to oxidation, which is prevented by α-TOH. As a result, α-TOH plasma level is decreased in obese individuals due to the consumption of α-TOH, as shown in several studies [72,73,74,131]. In obese compared to non-obese children, age-matched controls from the NHANES survey of α-TOH plasma level were 2.68 ± 0.59 vs. 3.17 ± 0.60 after the adjustment of TC and TG (*p* < 0.001) [75]. In contrast, other studies have reported plasma levels of α-, β-, γ-, and δ-TOH to be unchanged [76] or even increased (γ-TOH) in obese subjects [132]. Furthermore, α-TOH plasma level negatively correlates with waist circumference and waist-to-hip ratio [73], as well as body mass index (BMI), truncal fat mass, and total body fat mass [77]. A study by Verrotti et al. showed that epileptic girls treated with valproic acid for one year who became overweight had decreased α-TOH plasma levels, compared to baseline level or normal-weight adolescents [133]. The discontinuation of this drug [133] and weight loss [134] reversed obesity, and subsequently normalized antioxidant plasma levels. Adequate plasma concentrations of α-TOH are associated with reduced probability of overweight (OR: 0.56; 95% CI: 0.37, 0.86; *p* < 0.05) and obesity (OR: 0.38; 95% CI: 0.24, 0.60; *p* < 0.01) [77] and it is therefore important to prevent oxidative stress as an early event in the pathogenesis of complications such as DMT2 and CVD.

In summary, a large body of evidence shows that obesity is accompanied by increased oxidative stress resulting in low-grade inflammation. In obesity, the intake of fat and therefore of vitamin E is usually increased. Nevertheless, vitamin E plasma level is decreased in obese subjects, since vitamin E protects lipids from oxidation, and is therefore increasingly consumed. Consequently, obesity increases the risk of CVD and cardiovascular events. To offset the observed reduction in vitamin E level, supplementation of vitamin E in obese subjects seems to be a well-justified consideration. However, there is no evidence for this from any RCT (Figure 2).

#### 4.4.3. Diabetes Mellitus Type 2

Diabetes mellitus type 2 is the non-insulin-dependent form of diabetes mellitus, and is characterized by impaired function of pancreatic β-cells, resulting in insufficient insulin production, insulin resistance, and an increased plasma glucose level. An increased glucose plasma level correlates with increased oxidative stress and decreased vitamin E plasma concentration, possibly due to elevated phospholipid transfer protein (PLTP) level and activity [78]. PLTP transfers α-TOH from apolipoprotein B-containing lipoproteins to HDL-C or cellular membranes. Consequently, in patients with DMT2 and elevated plasma PLTP concentrations, vitamin E content in VLDL and LDL-C is reduced, which likely results in increased susceptibility of LDL-C to being oxidized and changes in vitamin E distribution [78]. Furthermore, Galvan et al. showed that insulin infusion decreased the vitamin E/LDL-C ratio by 10.0 ± 1.2% compared to saline infusion (*p* < 0.002), which suggests that insulin acts as a pro-oxidant agent, consuming vitamin E to buffer insulin-induced production of hydrogen peroxide [79]. The α-TOH and γ-TOH [135]-binding protein afamin in extravascular fluids or tissues was reported to be an independent predictor for the incidence of DMT2 in a meta-analysis of eight prospective cohort studies, with a total of 20,136 subjects [80]. An increase in afamin by 10 mg/L was associated with prevalent DMT2 diabetes (OR 1.19, 95% CI 1.12, 1.26) [80], whereas the association of afamin and vitamin E plasma level is still under debate.

Oxidative stress is accompanied by consumption of α-TOH; thus, the Insulin Resistance and Atherosclerosis Study (IRAS) found an inverse correlation between plasma α-TOH and diabetes incidence (OR 0.12, 95% CI 0.02, 0.68; *p* < 0.01) [81]. Diabetic patients with macroangiopathy had significantly lower α-TOH concentrations compared to those without vascular changes (*p* < 0.05), measured by N-acetyl-h-glucosaminidase activity and endothelial dysfunction [82]. Indeed, a 1 µmol/L decrease in plasma α-TOH levels was reported to increase diabetes risk by 22% (*p* = 0.0004) [136]. Individuals in the IRAS developing DMT2 had significantly lower α-TOH baseline levels compared to controls (25.1 ± 7.5 µmol/L vs. 28.1 ± 8.8 µmol/L; *p* < 0.01) [81]. Overall, reported differences in α-TOH plasma concentration are in the range of 28% [84] to 35% [85]. However, no correlation of nutritional vitamin E intake with incidence of diabetes was found [81].

Supplementation of 800 IU vitamin E per day for 4 weeks decreased the susceptibility of LDL-C to being oxidized [137]. Furthermore, α-TOH supplementation significantly decreased inflammatory level, measured by the levels of C-reactive protein (CRP) and IL-6, in patients with DMT2 compared to matched controls [138]. Combined supplementation of vitamin E (300 IU per day) and vitamin C (267 mg per day) for 3 months showed decreased blood glucose level and antioxidative capacity, measured by increased SOD and GSH enzyme activity, which lowered oxidative stress and consequently insulin resistance [86].

Obesity expedites insulin resistance, which, in the long term, results in impaired glucose tolerance, increased oxidative stress, and DMT2 [129]. In addition, absolute body fat content is inversely correlated with antioxidative status and α-TOH concentration, which itself inversely correlates with fasting plasma insulin concentration and the Homeostatic Model Assessment of Insulin Resistance (HOMA-IR) index [76]. A 3-month supplementation with 800 IU vitamin E per day in overweight individuals decreased fasting plasma glucose and improved insulin sensitivity, possibly independent of changes in inflammatory processes, which seemed not to be apparent after 6 months [87]. Other studies reported no correlation of increased insulin resistance in obesity with vitamin E concentration [73]. However, a meta-analysis of 14 RCTs involving 714 subjects revealed that vitamin E supplementation did not result in significant benefits to glycemic control as measured by reductions in hemoglobin A_1c_ (HbA_1c_), fasting glucose, and fasting insulin [88].

In conclusion, clinical trials show increased oxidative stress in patients suffering DMT2, which results in decreased vitamin E plasma levels. In addition, lower plasma α-TOH levels are correlated with an increased risk of development of diabetes. To the best of our knowledge, no study exists showing a correlation between nutritional vitamin E intake and the risk of developing diabetes. Although supplementation of vitamin E decreases oxidative stress and insulin resistance, insufficient evidence currently exists to support a potential beneficial effect of vitamin E supplementation in subjects with DMT2 [81] and related increased risk of CVD accompanying the progression and duration of the disease (Figure 2).

#### 4.4.4. Fatty Liver Disease

Non-alcoholic fatty liver disease (NAFLD) is globally the most common chronic liver disease [139], and often remains undiagnosed until a certain stage of the disease. Non-alcoholic fatty liver disease is a combination of inflammation, oxidative stress, and excessive accumulation of lipids in the liver. Inflammation in NAFLD is most likely accelerated by alteration of the gut microbial populations [140,141], and is hallmarked by activation of central inflammatory pathways such as NF-κB, signal transducer and activator of transcription (STAT)-3, and the inflammasome, resulting in increased production of pro-inflammatory cytokines. Similar findings have been reported for atherosclerosis and CVD [142,143]. During its development, several stages and severities of NAFLD can be defined [144], positively correlated with both the prevalence and the incidence of CVD [145]. Patients with non-alcoholic steatohepatitis (NASH), an advanced stage of fatty liver disease, appear to be at greater risk of CVD [145]. Since fatty liver disease and CVD share similar risk factors, such as DMT2, obesity, dyslipidemia, and hypertension, medications used to treat comorbidities are currently being used or tested for the therapy of NAFLD/NASH [146]. Dependent on the magnitude of the disease, lifestyle management and treatment with peroxisome proliferator-activated receptors (PPAR), γ agonists and vitamin E, more precisely α-TOH, have been shown to be effective therapeutic strategies. Fatty liver disease is an inflammatory disease, and thus elevated oxidative stress levels decreased α-TOH plasma levels in NASH patients compared to healthy subjects (22.4 vs. 26.8 nmol/mL; *p* < 0.01), whereas γ-TOH remained unchanged [89]. This observation supports the hypothesis that α-TOH is the most active antioxidative form of vitamin E. A recent study of Violet et al. showed alteration of α-TOH kinetics in women with obesity-associated hepatosteatosis compared to healthy controls, resulting in decreased release of α-TOH from the liver, consequently lowering the concentration of circulating α-TOH [91]. In addition, the authors postulate complex-like binding of α-TOH to lipids, which in the following avert α-TOH ROS-quenching properties and thus causing a worsening of the disease [91].

In contrast, other studies found increased α-TOH plasma levels in NASH patients (47.1 µM vs. 34.5 µM; *p* < 0.001; NASH vs. control) [90]. Nevertheless, vitamin E treatment was found to be one of the few established therapies for NAFLD and NASH, as shown in the Pioglitazone versus Vitamin E versus Placebo for the Treatment of Nondiabetic Patients with Nonalcoholic Steatohepatitis (PIVENS) study [92]. In this study, a two-year intervention with 800 IU/d α-TOH in non-diabetic patients with biopsy-diagnosed NAFLD significantly reduced the primary outcome (steatohepatitis) compared to placebo (43% vs. 19%; *p* = 0.001). α-Tocopherol plasma level was significantly associated with the genotype V433M of CYP4F2, the cytochrome that predominantly metabolizes vitamin E in the liver, as well as resolution of NASH and overall histological improvement [125]. However, these findings could not be replicated in the Treatment of Nonalcoholic Fatty Liver Disease in Children (TONIC) study in children diagnosed with obesity-induced NASH receiving 400 to 1200 IU α-TOH for up to 10 months [93] (Figure 2). Consequently, the relationship between CYP4F2 polymorphisms and the pharmacological effectiveness of vitamin E seems to be complicated by age differences [125].

In summary, inflammatory fatty liver disease is characterized by elevated oxidative stress resulting in decreased α-TOH plasma level and is also correlated with CVD. α-Tocopherol has been approved as an effective therapeutic strategy. However, polymorphisms of the hepatic vitamin E-metabolizing cytochrome are suggested to affect the pharmacological effectiveness of vitamin E supplementation.

#### 4.4.5. Metabolic Syndrome

Metabolic syndrome (MetS) typically combines five or more of the following metabolic disorders, namely, hyperlipidemia, hyperglycemia, hypertension, abdominal obesity, and insulin resistance [95,146]. Notably, each of these factors represents an increased risk for DMT2, NAFLD, and CVD, and thus, subjects suffering from one or more of these metabolic disorders possess an increased risk of developing these sequelae. Metabolic syndrome is characterized by increased oxidative stress and radical formation [95], consequently requiring a higher intake of antioxidants to avoid an imbalance between antioxidant levels and oxidative processes [97]. In addition, the bioavailability of vitamin E, more precisely α-TOH, was decreased in MetS patients compared to healthy subjects (12%; *p* < 0.05), possibly due to limited small-intestinal α-TOH absorption and/or impairment of hepatic α-TOH trafficking [97]. In fact, α-TOH plasma (18%, *p* = 0.042) [94,147] and organ level [148], as well as α-CEHC excretion level (41%, *p* = 0.002) [149], a biomarker for α-TOH status, are significantly lower in patients with MetS compared to non-MetS patients. Some studies show that α-TOH uptake and plasma levels are significantly higher in MetS patients [95]. However, there were no adjustments for lipid, cholesterol, or triglyceride uptake or plasma concentration of α-TOH, respectively [95,96]. Currently, an increased requirement for α-TOH in patients suffering from MetS is postulated, but not proven [97]. Indeed, a cohort of NHANES over 2001–2006 conducted in adults aged 20–85 years showed an inverse correlation between serum antioxidant status and MetS [95]. Although supplementation with α-TOH in rats reversed most of the metabolic disorders of MetS [150], no benefit of supplementation with α-TOH in a generally well-nourished population was observed, which is consistent with the outcomes for other diseases characterized by increased oxidative stress [151].

MetS combines several clinical characteristics associated with increased oxidative stress, resulting in an imbalance between antioxidant levels and oxidative processes. However, there is currently no intervention study in humans reporting beneficial effects of supplementation with α-TOH in a generally well-nourished population on the incidence of MetS and consequently CVD or cardiovascular events (Figure 2).

### 4.5. Cardiovascular Events, Particularly MI

An early study by Gey et al. [152] found a strong inverse association between plasma vitamin E level and IHD mortality. Furthermore, the risk of angina pectoris was inversely associated with the plasma concentration of vitamin E in a case-controlled population study of 110 cases of angina, even after adjustment for age, smoking habit, blood pressure, lipids, and relative weight [153].

Recently, Huang et al. reported in a long-term prospective cohort study, including biochemical analysis of 29,092 participants, that higher baseline serum α-tocopherol was associated with lower risk of overall mortality and mortality from all major causes. This study supports the long-term health benefits of higher serum α-TOH for overall and disease-specific mortality such as CVD [154]. Several observational studies [155,156,157,158,159,160,161,162] have consistently shown that vitamin E supplementation and/or high vitamin E intake is associated with a decreased risk of CVD. To our knowledge, only one Mendelian randomization study in China showed that high vitamin E levels were associated with an increased risk of CVD [163]. Despite this study, the overall consistency in the other studies has led many to suggest that vitamin E supplements may reduce the risk of CVD and several interventional trials have begun to study the cardioprotective effect of vitamin E.

Most studies have focused on vitamin E and the risk of CVD in general, while only a few have looked at the risk of major single causes of CVD like MI. A recent study from China stated that high vitamin E levels could increase the risk of MI [163]. A prospective study by Hak et al. [164] also reported that men without a history of CVD and with higher plasma vitamin E tended to have an increased MI risk. Hense and colleagues [165] found no association between serum vitamin E concentration and MI risk in their study population; however, they suggested that this might have been due to the high average levels of vitamin E in their study population.

A high plasma level may not be associated with a lower risk of MI; nevertheless, an interesting observation is a decrease in vitamin E plasma level in MI patients [166]. Within the first 48 h after MI, the plasma level of vitamin E declines significantly by 26% [167], and remains low until the third day after the start of the catabolic response [168]. Following an infarct, Sood et al. [169] showed that reperfusion was associated with excessive oxidative stress and increased consumption of this antioxidant not only in the ischemic but also in the reperfused myocardium. Vitamin E can be suggested as a valid marker for reperfusion and supplementation of vitamin E could be a therapeutic option for antioxidative protection of the myocardium in the acute setting.

Overall, numerous observational studies have consistently reported that high vitamin E intake or supplementation is associated with a decreased risk of CVD and overall mortality. However, no interventional trial in humans has shown, so far, the benefit of a supplementation of vitamin E to prevent any cardiovascular event. The decrease in vitamin E plasma level within the first 48 h after MI and the high demand for vitamin E during reperfusion might be promising therapeutic indications for short-term vitamin E supplementation.

## 5. Effects of Vitamin E in Chronic vs. Acute Events

The correlation of plasma level and supplementation of vitamin E with the incidence of cardiovascular events will be discussed in detail in the following chapter, and is summarized in Figure 3.

### 5.1. Atherosclerosis/Plaque Formation/Stability and Primary/Secondary Prevention

Several preclinical studies have demonstrated a preventive effect of vitamin E in plaque formation, such as preventing foam cell formation and endothelial dysfunction [170], scavenging free radicals in vascular SMC, preventing oxidative modification of LDL-C [171], reducing vascular SMC proliferation by protein kinase C [172,173,174,175,176], modulating endothelial cells [177], preventing the expression of adhesion molecules on endothelial cells [178,179] and mononuclear cell infiltration [180], adhering monocytes to endothelium [181], and curtailing the destabilization of fibrous plaques [182]. Furthermore, Schwenke and colleagues [183] showed, in rabbits fed with high doses of α-tocopherol, compared to low doses, significantly lower total plasma cholesterol and a decrease in atherosclerosis.

Despite these strong preclinical data on vitamin E’s effects on atherogenesis, clinical data has not supported these findings. Vitamin E supplementation does not attenuate plaque formation. The Vitamin E Atherosclerosis Prevention Study (VEAPS) [184] reported that, in healthy individuals at low risk of CVD, vitamin E supplementation did reduce LDL oxidation, but had no noticeable effect on the progression of atherosclerosis measured by carotid intima-media thickness (IMT). The Perth Carotid Ultrasound Disease Assessment Study (CUDAS) [185] demonstrated a progressive decrease in mean IMT with increasing quartiles of dietary vitamin E intake in men (*p* = 0.02) and a non-significant trend in women (*p* = 0.10). The Melbourne Atherosclerosis Vitamin E Trial (MAVET) [186] reported that vitamin E supplementation in chronic smokers failed to reduce the progression of carotid atherosclerosis. Furthermore, the Antioxidant Supplementation in Atherosclerosis Prevention (ASAP) trial, which combined vitamin E and C supplementation in hypercholesterolemic subjects [83], a randomized, controlled, double-blind trial, including 90 patients with coronary artery disease [187], and the Study to Evaluate Carotid Ultrasound Changes in Patients Treated with Ramipril and Vitamin E (SECURE) [188], did not observe differences in the progression of atherosclerosis between patients with vitamin E supplementation and placebo.

Overall, the preventive effects of vitamin E alone or in combination with vitamin C seen in preclinical models of atherosclerosis have not been confirmed in clinical trials. Long-term supplementation of vitamin E does not seem to attenuate atherogenesis in humans (Figure 3).

### 5.2. Myocardial Infarction

Most studies have focused on the benefits of vitamin E supplementation in CVD, as reported above in Section 4.5. However, a few primary and secondary prevention trials have looked more specifically into whether vitamin E supplementation can reduce the incidence of fatal or non-fatal MI.

First, the primary prevention of angina pectoris by α-TOH supplementation in healthy smokers aged 50–69 years was studied in the Finnish Alpha-Tocopherol Beta Carotene (ATBC) study [189]. α-Tocopherol supplementation was associated with only a minor decrease in the incidence of angina pectoris. A further study, the Collaborative Primary Prevention Project (PPP) [190], investigated the efficacy of vitamin E (α-tocopherol) in primary prevention of cardiovascular events in people with one or more major cardiovascular risk factor. Vitamin E supplementation showed no effect on CVD death, total cardiovascular events, or MI. Thus, long-term, high-dose vitamin E supplementation fails to prevent MI.

Several trials were performed investigating secondary prevention of cardiovascular events by vitamin E supplementation in patients with clinical evidence of CVD. In the Cambridge Heart Antioxidant Study (CHAOS) [191], supplementation with vitamin E substantially reduced the incidence of non-fatal MI after 1 year of treatment. A further study, the Secondary Prevention with Antioxidants of Cardiovascular Disease in Endstage Renal Disease (SPACE) trial [192], reported a significant decrease in acute MI in hemodialyzed patients with vitamin E supplementation. However, a further large prospective trial, the Gruppo Italiano per lo Studio della Supervienza nell’Infarto miocardico (GISSI), including 11,324 subjects with a recent MI, showed no effect of vitamin E treatment on non-fatal MI [193]. Notably in this study, all subjects were invited to follow a Mediterranean diet, which may have led to a high vitamin E food intake in all study groups. Furthermore, a meta-analysis of nine studies with 80,645 participants reported an association between vitamin E supplementation and a reduction in non-fatal MI in patients with pre-existing coronary artery disease, but no association with a reduction in total mortality or total CVD mortality [194].

The secondary prevention of MI with vitamin E supplementation is controversial and might be a question of treatment strategy and patient profiling. Moreover, the presented trials have tested vitamin E’s capacity to provide plaque stabilization, reduce the risk of plaque rupture, or prevent cardiovascular events like MI, but not the potential of vitamin E in preserving cardiac function in the event of acute MI. There are only limited data addressing this question. Nevertheless, the preclinical studies available show potential benefits of vitamin E in models of ischemia/reperfusion injury in various organs, including a few early studies on cardiac ischemia/reperfusion injury [195,196,197,198]. The first clinical trials have also shown promising results. The Myocardial Infarction and VITamins (MIVIT) trial [199] showed in a pilot trial including 800 patients a positive influence on the clinical outcome for patients with acute MI, as well as the Indian Experiment on Infarct Survival-3 [200]. This study suggested that combined treatment with antioxidant vitamins A, E, C, and β-carotene in patients with recent acute MI improved clinical outcomes and may have been protective against cardiac necrosis and oxidative stress. To our knowledge, there are only a few pilot studies looking at cardioprotection in the acute setting of MI, and in these trials, vitamin E supplementation is combined with other antioxidants. The effect of vitamin E supplementation alone on infarct size and preservation of cardiac function in MI patients has not been investigated so far. Therefore, clinical studies are warranted to investigate the potential of vitamin E medication in the acute setting of MI.

In summary, primary and secondary prevention of MI with vitamin E supplementation has largely failed to prevent MI. Vitamin E treatment directly in patients presenting with MI with the aim of preventing acute cardiac ischemia/reperfusion injury might be a promising vitamin E medication strategy where further clinical trials are warranted (Figure 3).

## 6. Controversial Outcome of RCTs—Explanatory Approaches

As reported earlier, studies investigating vitamin E’s potential to reduce cardiovascular events have failed to show consistent results [201]. The potential reasons for this are diverse but important to understand. The potential impacts of proteins regulating vitamin E homeostasis [202], different doses and durations of vitamin E supplementation, as well as the specific vitamin E form applied, have been widely discussed. Furthermore, interactions of the different vitamin E forms with each other or with co-treated medications must also be taken into consideration (Figure 4).

### 6.1. α-TOH vs Non-α-TOH Forms

In interventional studies, α-TOH is the most common form used to investigate the cardiovascular effects of vitamin E, due to its known anti-inflammatory effects, in addition to its property as an antioxidant. However, whether α-TOH is effective when given without other vitamin E forms is not clear. In observational studies, the possible beneficial effects of non-α-TOH forms, namely β-, γ-, δ-TOH and α-, β-, γ-, δ-T3, cannot be separated from those of α-TOH.

After α-TOH, γ-TOH and γ-T3 are the next most interesting forms of vitamin E, due to their potent antioxidative and anti-inflammatory capacity. Notably, γ-TOH is the only form of vitamin E characterized by an unsubstituted C-5 position, thus enabling electrophile trapping, which results in the detoxification of NO_2_ and peroxynitrite via the formation of 5-nitro-γ-TOH [203]. Supplementation with γ-TOH-rich mixed TOHs showed enhanced inhibition of inflammatory status, as assessed by decreased plasma levels of pro-inflammatory markers, such as CRP, IL-6 [204,205,206], and F2-isoprostane, and reduced leukotriene B_4_ from stimulated neutrophils [207,208]. As reviewed by Mathur et al., supplementation with γ-TOH affected markers of atherosclerosis in humans by attenuating oxidative and nitrosative stress via improvement of vascular endothelial function and lipid peroxidation, as well as coagulation and platelet aggregation [209].

The anti-inflammatory properties of non-α-TOH forms, particularly T3, indicate beneficial effects in the prevention and therapy of chronic diseases [210]. Qureshi and colleagues are the pioneers in T3 research, describing protective effects in relation to risk factors for the progression of atherosclerosis and CVD in hypercholesterolemic humans, such as the lowering of serum cholesterol [211,212]. These findings were supported by a study by Yuen et al., showing significant decreases in total cholesterol and LDL-C plasma concentrations after 5 months of supplementation with 300 mg per day mixed T3s [213]. Furthermore, the administration of T3 at 100 and 200 mg per day for 2 months improved arterial compliance in healthy men [214]. However, large-scale human trials investigating the cardioprotective effects of non-α-TOH forms are still to be performed.

It must be considered that the occurrence of T3 is rare and its bioavailability in humans is less than 1% compared to α-TOH, and thus, an increase in nutritional T3 is not automatically associated with a significant increase in circulation. In addition, non-α-TOH forms are typically rapidly metabolized. Consequently, plasma concentrations of the non-α-TOH forms are decreased and respective metabolites can be formed. Therefore, the beneficial effects of non-α-TOH forms on atherosclerosis and related cardiovascular events may be missed. Furthermore, these metabolites independently mediate anti-inflammatory and antioxidative properties [210].

In summary, the different strengths of the antioxidative and anti-inflammatory properties of non-α-TOH forms of vitamin E compared to α-TOH are known. Based on the current state of research, γ-TOH and T3, as well as hepatically formed metabolites of α- and γ-TOH, are promising forms of vitamin E in the prevention of diseases driven by acute inflammatory and oxidative processes such as MI. To verify this, large-scale clinical trials are needed.

### 6.2. Relevance of Metabolites

Within the hepatically formed metabolites of vitamin E, SCMs as metabolic end-products were first described in relation to anti-proliferative and anti-inflammatory properties in cancer and immune cells, respectively [215]. In addition, the LCM of vitamin E, which are the first hepatic products formed during the metabolism of vitamin E, are described to make valuable contributions to the biological activity of vitamin E. Long-chain metabolites and their precursors, tocopherols and tocotrienols, differ only in the terminal oxidation of the side-chain, and thus, similar modes of action can be assumed. Birringer et al. allocated the substructures of different forms and metabolites of vitamin E to specific functional moieties, assigning the LCM, with a terminal hydroxyl and carboxyl group at the side-chain, as structures with higher anti-inflammatory activity than their parent chromanols and chromenols [216]. Indeed, research on these metabolites has revealed the effects of LCM on inflammatory processes [217,218,219,220] and lipid metabolism [221,222], thus providing evidence of their physiological relevance [215]. A recent study by our group investigating the effect of δ-T3-13′-COOH on the progression of atherosclerosis showed protective effects on intra-plaque inflammation, as measured by nitrotyrosine [219]. Furthermore, direct and indirect effects of LCM on the inflammasome NLRP3, which has been reported to play a key role in the development of atherosclerosis, have been discussed [144]. However, no effect on the formation of atherosclerotic plaques in high-fat diet fed Apoe^−/−^ mice was observed [219]. Data available on the mode of action of LCM revealed that 13′-OH and 13′-COOH, at least in part, mediated different pathways in immune cells, as known from α-TOH, and mediated the observed effects in lower concentrations, as observed for α-TOH. Taking this into account, the effects of vitamin E seem to be complicated by the circulating hepatically formed LCM of vitamin E [223]. Therefore, we are confident that LCM need to be considered in evaluation of the role of vitamin E in CVD, and as potential leading structures for drug development for the treatment of CVD as well as NAFLD/NASH [144].

The biological mode of action of the LCM and their relevance in human trials need to be investigated further. To date, RCTs investigating the effects of LCM on atherosclerosis and the cardiovascular event rate are lacking.

### 6.3. Comparison of Vitamin E and Anti-Inflammatory Therapies

Different forms of vitamin E are beneficial agents in protecting against oxidative and inflammatory stress. However, the effects reported in human trials are rather inconsistent, as shown for α-TOH, or large-scale human trials are missing, as described for non-α-TOH forms, making it difficult to reach a conclusion on the beneficial effects of vitamin E in primary and secondary prevention of cardiovascular events.

In contrast, recent large-scale clinical trials using highly specific anti-inflammatory therapies, such as an IL-1β-neutralizing antibody, methotrexate, or colchicine, have revealed, at least in part, a reduction in cardiovascular events, thus, notably confirming the major contribution of inflammation to CVD, specifically atherosclerosis. Among the central therapeutic targets are pro-inflammatory cytokines, with a special focus on IL-1β. Interleukin-1β is mainly activated by the inflammasome nucleotide-binding domain and leucine-rich repeat pyrin domain (NLRP) 3, which has been shown to play a major role in atherosclerosis and CVD [224]. A recent review showed a blocking effect of vitamin E on the priming and activation process of the NLRP3 inflammasome and the related formation of IL-1β [144].

In the Canakinumab Antiinflammatory Thrombosis Outcome Study (CANTOS), patients with elevated CRP level ≥2 mg/L were randomized in four groups receiving none, 50 mg, 150 mg, and 300 mg IL-1β neutralizing antibody every 3 months. Targeting IL-1β revealed a decrease in CRP concentration of 26% to 41% after 4 years of treatment without affecting the plasma lipid profile [225]. Elevated CRP plasma levels are associated with peripheral vascular disease, coronary artery disease, MI, and stroke [226]. The first occurrence of a non-fatal MI, any nonfatal stroke, and cardiovascular death were defined as the primary endpoints. Regarding these primary endpoints, treatment with 150 mg and 300 mg revealed HR for the primary IL-1β neutralizing antibody with 15% and 14% risk reduction, respectively [225]. However, re-analysis of the CANTOS trial further showed that patients receiving canakinumab were still at increased risk of recurrent cardiovascular events, possibly mediated by IL-18 and IL-6, which are also potential targets for future treatment strategy for CVD [227]. Taking this into account, targeting multiple signaling pathways and the formation of cytokines could be promising strategies in the treatment of CVD. This raises the question of whether vitamin E, as a broad-range anti-inflammatory agent, is still to be considered a promising intervention in CVD and atherothrombosis.

In contrast to the CANTOS trial, the clinical Cardiovascular Inflammation Reduction Trial (CIRT) targeted a broader range of pro-inflammatory processes using the anti-rheumatic compound methotrexate, which is known to reduce cytokine secretion and the generation of ROS [228]. After the primary prevention of rheumatoid arthritis, off-target effects such as a reduction in cardiovascular events were detected with the methotrexate treatment using the normal dosing range of 5–30 mg per week for rheumatoid arthritis [229]. In earlier studies, methotrexate was associated with a 21% risk reduction in CVD (*p* < 0.001) and an 18% lower risk of MI (*p* = 0.01) among patients with rheumatoid arthritis, psoriasis, or polyarthritis [230]. However, a recent clinical trial by Ridker et al. enrolled patients with previous MI or multivessel coronary disease and additional DMT2 or MetS; after a total study duration of 8 months applying methotrexate at 15 mg per week, followed by 20 mg per week, clinical parameters were assessed. Neither any markers of inflammation, such as IL-6, IL-1β, or CRP, nor the cardiovascular event rate were altered by methotrexate [231]. The differences in outcomes of this methotrexate trial compared with the above-described trial with canakinumab could be the result of different inflammatory disease stages in the enrolled patients. Beneficial effects of methotrexate were observed in patients suffering from rheumatoid arthritis, psoriasis, or polyarthritis, which are associated with a certain level of inflammation, whereas in the CIRT patients diagnosed with DMT2 and MetS were enrolled. Since the degree of inflammation is a key player in CVD, the study outcomes may have been altered due to this difference [144,225]. A beneficial effect of methotrexate on the cardiovascular event rate seems to be present in arthritis patients, whereas in others with a less inflammatory status beneficial effects may not be expected.

Another approach in the prevention of cardiovascular events is the drug colchicine, which has long been in use for the treatment of gout and pericarditis. Among other properties, colchicine attenuates the production of inflammatory chemokines and the activation of the inflammasome NLRP3 [232,233]. The first clinical trial evaluating the effects of low-dose colchicine (LoDoCo) in patients with stable coronary disease revealed a significant reduction in cardiovascular events (HR 0.33; 95% CI 0.18, 0.59; *p* < 0.001) in patients supplemented with 0.5 mg colchicine per day for 3 years compered to non-supplemented patients [234]. The recent Colchicine Cardiovascular Outcomes Trial (COLCOT) investigated the efficacy of the same dose of colchicine for 2 years after the occurrence of an MI [235]. The HR for a defined composite of primary endpoints, death from cardiovascular causes, resuscitated cardiac arrest, MI, stroke, or urgent hospitalization for angina leading to coronary revascularization, decreased to 0.77 (95% CI 0.61, 0.96; *p* = 0.02). However, single HR for primary endpoints showed decreasing trends for death from cardiovascular causes (HR 0.84, 95% CI 0.46, 1.52) and MI (HR 0.91, 95% CI 0.68, 1.21) [235].

Taken together, we can conclude that the success of anti-inflammatory therapies seems to be dependent on the initial status of oxidative and inflammatory stress, as measured by an increased CRP level. Subjects with persistently elevated inflammatory states seem to benefit most from therapeutic strategies targeting inflammation regarding the related risk of CVD.

### 6.4. Importance of Cohort Selection and Hp 2-2 Genotype

The protein haptoglobin (Hp) circulates in the blood and acts as an antioxidant which binds free hemoglobin and thus protects tissues from iron-induced oxidation [236]. In humans, the homozygous genotypes Hp 1-1 and Hp 2-2 and the heterozygous genotypes Hp 2-1 and Hp 2-1m exist, with Hp 2-2 being the most common genotype [237,238]. Compared to Hp 1-1 and Hp 1-2, Hp 2-2 is considered an independent risk factor for cardiovascular outcomes among individuals with DMT2 [239]. The analysis of the Strong Heart Study (SHS) in 4549 Native American showed the following susceptibilities to CVD in DMT2 patients: Hp 1-1 (OR 1.0) < Hp 1-2 (OR 1.63) < Hp 2-2 (OR 4.96; *p* = 0.002) [239]. A similar trend was present in non-diabetic subjects, which supports the hypothesis that Hp 1-1 offers superior antioxidant protection [236]. Furthermore, Hp 2-2 is associated with an increased level of CRP and carotid arterial IMT, a marker for CVD, in a cohort from Singapore [238] and the Diabetes Heart Study conducted in European Americans [240]. However, the study of De Bacquer et al. reported an increased risk of cardiovascular events in non-diabetic patients with the Hp 1-1 phenotype in the Belgian Interuniversity Research on Nutrition and Health (BIRNH) survey [241]. The relevance of the Hp genotype for the cardiovascular risk in patients suffering DMT1 remains elusive [242].

Different Hp genotypes are associated with different oxidative conditions; thus, proper sample collection is crucial in human studies investigating the cardioprotective effects of vitamin E. A meta-analysis of the Heart Outcomes Prevention Evaluation (HOPE) trial and the Israel Cardiovascular Events Reduction with Vitamin E (ICARE) study revealed significant reductions in cardiovascular death, MI, and stroke (OR 0.58, 95% CI 0.40, 0.86; *p* = 0.006) in diabetic individuals with the Hp 2-2 genotype supplemented with vitamin E compared to diabetics with the Hp 1-1 or Hp 2-1 genotype [243]. Further, including the Woman’s Health Study (WHS) in this analysis, the odds ratio was 0.66 (95% CI 0.48, 0.90; *p* = 0.009) [244]. A recent study by Asleh et al. confirmed a reduction in cardiovascular events in patients with DMT2 and the Hp 2-2 genotype (OR: 0.66, 95% CI 0.45, 0.95; *p* = 0.025) via high-dose vitamin E supplementation [245]. In contrast, in subjects with non–Hp 2-2 genotypes, no beneficial effects of vitamin E supplementation on the cardiovascular event rate has been observed.

Beyond the Hp2-2 genotype, other genotypes may affect the outcome of vitamin E supplementation. Several studies demonstrated an impact of e.g., apolipoprotein E (ApoE) genotype [246,247,248] and Paraoxonase 1 (PON-1) genotype [249,250] on vitamin E. For example, the apoE4 genotype represents a significant risk factor for CVD. ApoE4 may be associated with lower vitamin E retention in peripheral tissues, possibly related to a combination of several factors: An increased α-TOH retention in LDL, an impaired cellular vitamin E delivery system, a reduced expression of lipoprotein receptors, and an increased intracellular degradation of vitamin E [248].

In summary, the Hp 2-2 genotype is associated with increased oxidative stress and an increased risk of cardiovascular events, especially in diabetic patients. Therefore, vitamin E supplementation in this sub-cohort of patients seems to be promising in protecting from CVD. Re-analyzing previous RCTs with respect to Hp genotypes could provide insights into the differences in the beneficial effects of vitamin E supplementation.

### 6.5. High-Dose Supplementation of Vitamin E vs. Balancing Vitamin E Deficiency

Long-term supplementation of vitamin E to prevent cardiovascular events has been intensively studied and reported to be inefficient in many human intervention and correlation studies. Various doses of vitamin E, ranging from 16.5 to 2000 IU per day have been tested. A meta-analysis including 135,967 objects in 19 clinical trials investigated the relationship between vitamin E dosage and all-cause mortality. Eleven trials supplemented with high-dosage vitamin E (≥400 IU per day), and nine of these 11 trials showed an increased risk for all-cause mortality [29]. Long-term supplementation of high-dosage vitamin E is therefore not recommended.

Only specific risk groups may need a higher vitamin E intake to receive the same vitamin E plasma level as healthy subjects. For example, patients with MetS showed higher requirements, due to greater inflammation and oxidative stress that limit absorption and/or impair hepatic α-tocopherol trafficking [251]. However, this higher intake does not mean dosages greater than the FDA-approved dose of 400 IU per day, which is considered safe in humans without adverse effects.

In contrast, special conditions like acute MI may lead to higher demand for vitamin E. As mentioned in Section 4.5, vitamin E plasma levels drop within the first 48 h after acute MI and therapeutic reperfusion also leads to increased consumption of vitamin E. Even low doses of vitamin E, delivered at the crucial time around reperfusion to meet the higher vitamin E demand, have been reported to be highly effective and cardioprotective in the preclinical setting of an acute MI [195]. Thus, the duration and time of supplementation, as well as the dosage, of vitamin E could be crucial for the beneficial or deleterious effects of vitamin E therapy.

Overall, long-term supplementation of vitamin E fails to prevent cardiovascular events, as shown in many correlation and intervention studies, and high-dose vitamin E supplementation is not recommended. Supplementation of vitamin E in conditions of higher demand like the setting of an acute MI and during reperfusion could be a beneficial therapeutic strategy.

## 7. Merits and Limitations

The focus in this review is on the effects of vitamin E in association with the risk factors for CVD and CVD itself, particularly MI. As such, we summarize the relationships between vitamin E levels and various risk factors for CVD, and discuss potential reasons why the association of low vitamin E levels and cardiovascular risk found in observational studies could not be transferred to successful treatment options in interventional studies. We hypothesize that vitamin E application in the acute treatment of myocardial infarction is likely more successful than in chronic supplementation aiming at the prevention of cardiovascular events.

We decided to exclude stroke, as it is an often discussed, controversial, and extensive field itself, which goes beyond our review. For more interest in stroke, we refer readers to other reviews and meta-analyses focusing on stroke and vitamin E [252,253,254,255,256].

## 8. Conclusions

Overall, extensive studies have tested the potential of vitamin E to prevent cardiovascular events, in particular MI. Promising data from observational studies have reported associations between higher vitamin E intake and higher vitamin E plasma levels and lower risk of cardiovascular events, as well as promising associations between vitamin E intake/plasma levels and risk factors for CVD, such as hypertension, hyperlipidemia, venous thrombosis, DMT2, NAFLD/NASH, age, obesity, and metabolic syndrome. These effects have not been confirmed in interventional trials assessing the impact of vitamin E supplementation on the CVD risk factors hyperlipidemia, hypertension, obesity, and metabolic syndrome, as well as on CVD outcomes. A few interventional trials assessing the effect on vitamin E supplementation on venous thrombosis showed controversial results and, in our opinion, more studies in this field are required. Positive effects of vitamin E supplementation could be observed in adult NAFLD/NASH patients, as well as in DMT2 patients. Further RCTs for DMT2 patients and vitamin E supplementation are warranted. Overall, many clinical trials have presented controversial results, suggesting that long-term vitamin E supplementation is not a favorable therapeutic strategy, and that high-dose vitamin E supplementation is not to be recommended. Furthermore, the results from vitamin E interventional trials also seem to be highly controversial, mainly due to the wide range of different parameters studied. Critical parameters for RCTs are cohort selection, duration, timing and dosage of vitamin E treatment, the different forms of vitamin E, the synergistic effects of vitamin E, and the metabolism of vitamin E. These parameters need to be critically reviewed based on the outcome of previous trials in the specific clinical setting and need to be thoroughly assessed and considered for future and hopefully more successful study designs.

Vitamin E therapy in the setting of an acute cardiac ischemic event and during reperfusion, when the vitamin E plasma level is down and the demand for vitamin E is increased, has shown promising preclinical effects on infarct size and preservation of cardiac function. The success of vitamin E therapy could potentially be significantly improved by treatment in the acute setting, along with vitamin E deficiency.

## Figures and Tables

**Figure 1 antioxidants-09-00935-f001:**
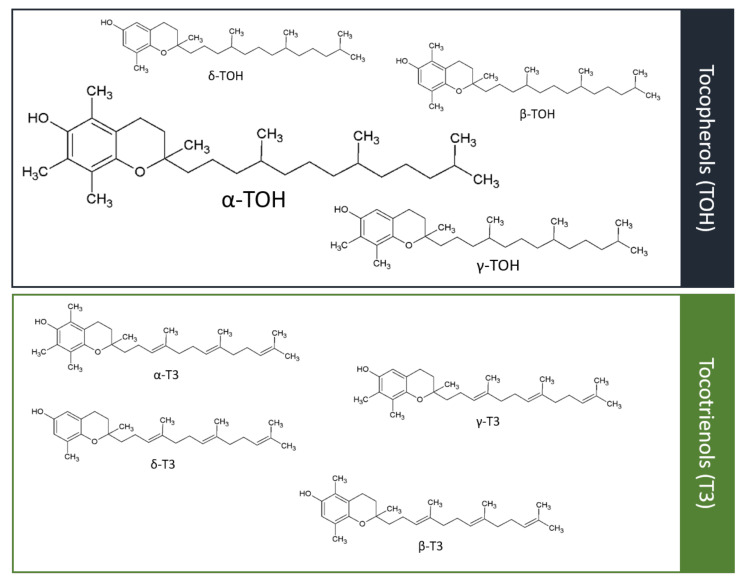
Different forms of vitamin E. The vitamin E group consists of eight different forms, α-, β-, γ-, δ-tocopherol (TOH) and their respective tocotrienols (T3), α-TOH being the most prominent form in human nutrition and in the human body. Other forms of vitamin E, such as tocomonoenols, are known, but are not relevant in human nutrition, and are therefore not addressed in this review.

**Figure 2 antioxidants-09-00935-f002:**
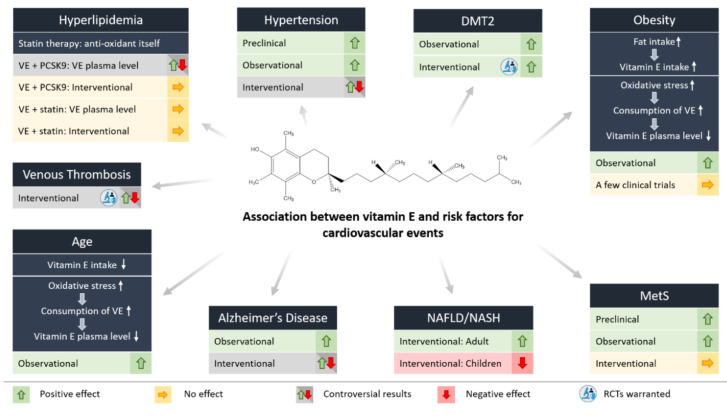
Association between vitamin E and risk factors for cardiovascular events. Correlations between vitamin E plasma level and effects of vitamin E supplementation on single risk factors for cardiovascular events are shown. Risk factors presented here are either dependent (hyperlipidemia, diabetes mellitus type 2 (DMT2), obesity, age, Alzheimer’s disease, non-alcoholic fatty liver disease (NAFLD), non-alcoholic steatohepatitis (NASH), metabolic syndrome (MetS)) or independent (hypertension, venous thrombosis) of oxidative processes and inflammation. Most of the risk factors are positively affected by vitamin E, showing the beneficial potential of vitamin E in cardiovascular events. However, randomized clinical trials have revealed controversial results or no effect. Abbreviations used in the figure: vitamin E, VE.

**Figure 3 antioxidants-09-00935-f003:**
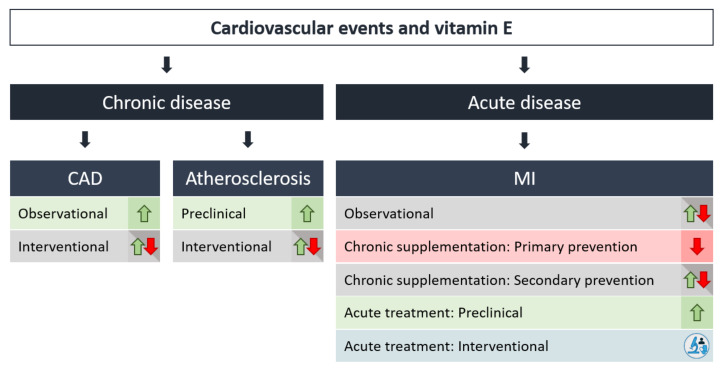
Correlation of plasma level and supplementation of vitamin E with incidence of cardiovascular events. Relevance of vitamin E to the risk of cardiovascular diseases (CVD) depends on the inflammation grade resulting from chronic and acute cardiovascular events. Increased vitamin E plasma levels correlate with decreased risk of chronic CVD, ischemic heart disease (IHD), and atherosclerosis, whereas supplementation of vitamin E has revealed controversial results. In acute cardiovascular events, such as myocardial infarction (MI), acute treatment of vitamin E is most promising in balancing a deficiency of antioxidants. Abbreviations used in the figure: vitamin E, VE.

**Figure 4 antioxidants-09-00935-f004:**
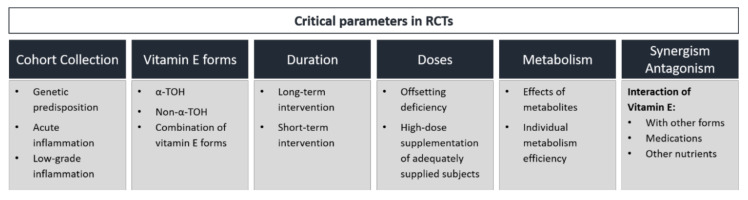
Critical parameters in randomized controlled trials (RCTs) investigating beneficial effects of vitamin E supplementation on CVD. Vitamin E is the most prominent lipid-soluble antioxidant additionally possessing anti-inflammatory capacity. However, in randomized controlled trials (RCTs) supplementation of vitamin E has revealed controversial effects on the risk of cardiovascular events. The reasons for these findings are diverse. The most common reasons discussed are: cohort selection, the form of vitamin E used for treatment and the treatment durations and doses of vitamin E, the relevance of hepatically formed metabolites, and the synergistic or antagonistic effects of different vitamin E forms on each other or on nutritional factors and medications.

**Table 1 antioxidants-09-00935-t001:** Vitamin E and Risk Factors for Cardiovascular Events.

Risk Factor	Type of Study	Author	Participants	Endpoints	Vitamin E Dosage
**Hypertension**					
SBP (systolic blood pressure) DPB (diastolic blood pressure)	Observational	Kuwabara et al. [53]	*n* = 3507	higher vitamin E intake is associated with a lower percentage of subjects with hypertension	
	Interventional	Boshtam et al. [54]	*n* = 70 mild hypertensive patients	Significant decrease in SBP and DBP (mainly in SBP)	134 mg per day (200 IU) for 27 weeks
	Interventional	Tmj et al. [55]	*n* = 60 mild hypertensive subjects	decrease in blood pressure	134 mg (200 IU) per day for 12 weeks
	Interventional	Palumbo et al. [56]	*n* = 142 treated hypertensive patients	no clinically relevant effect on blood pressure	300 mg per day for 12 weeks
	Interventional	Mihalj et al. [57]	*n* = 57 treated hypertensive patients	no further effect of vitamin E/C supplementation	720 mg vitamin E and 25 mg vitamin C per day for 8 weeks
	Interventional	Barbagallo et al. [58]	*n* = 12 hypertensive patients	no effect of vitamin E treatment on SBP or DBP	600 mg vitamin E per day for 4 weeks
**Hyperlipidemia**					
Hypercholesterolemic (HC)	Observational Interventional	Cangemi et al. [59]	*n* = 30 HC patients *n* = 20 healthy subjects *n* = 30 HC patients	lower vitamin E plasma level in HC patients vs. healthy subjects administration of atorvastatin restored vitamin E/TC plasma level	10 mg atorvastatin per day for 30 days
	Observational	Shin et al. [60]	*n* = 76 HC patients	increased α-TOH/lipid plasma level in HC patients	20–40 mg simvastatin per day for 8 weeks
	Observational	Blom et al. [61]	*n* = 738 HC patients	increased vitamin E/TC plasma level in evolocumab (anti- PCSK9 antibody)-treated patients from baseline to week 52	10 or 80 mg atorvastatin per day or 80 mg atorvastatin plus 10 mg ezetimibe per day for 52 weeks, 420 mg Evolocumab or placebo for 8 weeks
	Interventional	Lui et al. [62]	*n* = 19 HC patients	increased vitamin E/LDL-C plasma level in atorvastatin-treated HC patients	10 mg atorvastatin per day for 5 months
	Interventional	Leonard et al. [63]	*n* = 44 HC patients	vitamin E supplementation did not alter cholesterol levels under statin therapy	268 mg (400 IU) vitamin E per day or placebo for 12 weeks
**Thrombosis**					
	Interventional	Glynn et al. [64]	*n* = 39,876 women aged 45 and older	women taking vitamin E were 21% less likely to suffer a venous thromboembolism	Vitamin E (540 mg) or a placebo on alternate days over a 10-year period.
	Interventional	Vuckovic et al. [65]	2506 patients with venous thrombosis, 2506 partner controls, and 2684 random-digit-dialing (RDD) controls *n* = 96 patients supplemented with vitamin E	No association of vitamin E supplementation with a reduced venous thrombosis risk	No information was obtained on the dosage of vitamin E intake
**Age**					
	Observational	Ortega et al. [66]	*n*= 120 aged subjects (65–91 years)	Lower vitamin E intake and α-TOH/TC plasma level correlates with cognitive impairment in elderly	-
	Observational	Vatassery et al. [67]	48 healthy male volunteers aged 24–91 years	α-TOH plasma level remained unchanged, decreased α-TOH level in platelets of elderly subjects	-
	Observational	Capuron et al. [68]	*n* = 69 aged subjects (73–86 years)	Lower α-TOH plasma level in subjects with poor physical and mental health status	-
	Observational	Requejo et al. [69]	*n* = 120 aged subjects (65–91 years)	95.2% are below recommendations of a-TOH intake	-
	Observational	Rudman et al. [70]	*n* = 34 eating-dependent nursing home residents	The vast majority did not receive micronutrient supplements	-
	Interventional	De la Fuente et al. [71]	*n* = 33 aged subjects (65–75 years) *n*= 30 controls (25–35 years)	α-TOH improves immune functions and therefore health in aged people	200 mg α-TOH per day for 3 months
**Obesity**					
	Observational	Silva et al. [72]	*n* = 33 overweight adolescents *n* = 42 obese adolescents *n* = 75 healthy adolescents (10–15 years)	Crude and energy-adjusted intake of vitamin E positively correlate with BMI, but not with plasma level of vitamin E; α-TOH/LDL-C and α-TOH/TC decrease in obese and overweight adolescents	-
	Observational	Mehmetoglu et al. [73]	*n* = 98 obese patients *n* = 78 healthy subjects (18–65 years)	decreased α-TOH/TC + TG plasma level in obese subjects	-
	Observational	Kljno et al. [74]	*n* = 17 obese girls *n* = 7 healthy girls (8–15 years)	α-TOH/total lipids decreased in plasma and in LDL in obese subjects	-
	Observational	Strauss et al. [75]	*n* = 6139 children (6–19 years) enrolled in the NHANES III	decreased α-TOH/TC + TG plasma level in obese subjects	-
	Observational	Molnar et al. [76]	*n* = 15 obese adolescents *n* = 16 healthy adolescents (13–16 years)	α-TOH/TC + TG plasma level remained unchanged in obese subjects	-
	Observational	Gunanti et al. [77]	6139 children (8–15 years) enrolled in the 2001–2004 NHANES	Adequate plasma level of α-TOH/TC are associated with reduced probability of overweight	-
**DMT2**					
DMT2 (Diabetes mellitus type 2)	Observational	Schneider et al. [78]	*n* = 31 DMT2 patients (46–79 years) *n* = 31 control subjects (38–63 years)	VLDLs and LDLs of DMT2 patients contained fewer vitamin E molecules compared to controls due to PLPT	-
	Observational	Galvan et al. [79]	*n* = 12 male DMT2 patients (49–54 years) *n* = 19 control subjects (29–34 years)	Insulin infusion decreased α-TOH/LDL-C plasma level	-
	Observational (meta-analysis)	Kollerits et al. [80]	*n* = 20,136 subjects	Vitamin E-binding protein afamin is an independent predictor for DMT2 incidence, increase in afamin is associated with prevalence DMT2	-
	Observational/ Interventional	Mayer-Davis et al. [81]	*n* = 895 non-diabetic adults (45–65 years) (*n* = 318 non-supplement users and *n* = 577 supplement users)	α-TOH plasma level is decreased in DMT2 patients and correlates with diabetes incidence, but not the nutritional intake/use of supplements	-/not defined
	Observational	Škrha et al. [82]	*n* = 62 DMT2 patients (49–64 years) *n* = 20 controls subjects	decreased α-TOH/TC, TG serum level in diabetic patients with macroangiopathy versus without vascular changes	-
	Observational	Salonen et al. [83]	*n* = 944 male healthy subjects (42–60 years)	decreased α-TOH plasma levels associated with increase diabetes risk	-
	Observational	Eljaoudi et al. [84]	*n* = 60 DMT2 patients *n* = 40 healthy subjects (31–76 years)	decreased α-TOH plasma level in DMT2	-
	Observational	Nourooz-Zadeh et al. [85]	*n* = 87 DMT2 patients *n* = 41 healthy subjects (17–86 years)	decreased α-TOH/TC plasma level in DMT2	-
	Observational	Mehmetoglu et al. [73]	*n*= 98 obese subjects *n* = 78 healthy subjects (18–65 years)	no correlation of α-TOH/TC + TG plasma level and insulin resistance in obese subjects	-
	Interventional	Rafighi et al. [86]	*n* = 170 DMT2 patients (30–60 years)	Vitamin E supplementation decreased blood glucose level, antioxidative capacity, (increased SOD and GSH enzyme activity), oxidative stress and insulin resistance	200 mg (300 IU) vitamin E (/day) and 267 mg vitamin C per day for 3 months
	Interventional	Manning et al. [87]	*n* = 80 healthy subjects (38–57 years)	Vitamin E supplementation decreased inflammatory processes, fasting plasma glucose and improved insulin sensitivity in overweight subjects	537 mg (800 IU) vitamin E per day or placebo for 3 months
	Interventional (Meta-analysis)	Xu et al. [88]	*n* = 714 subjects	vitamin E supplementation did not change glycemic control (HbA1c, fasting glucose, fasting insulin)	134–1074 mg (200–1600 IU) per day for 6–27 weeks
**Fatty Liver Disease**					
	Observational	Erhardt et al. [89]	*n* = 50 NASH patients *n* = 40 healthy controls (35–67 years)	decreased α-TOH plasma levels in NASH patients	-
	Observational	Machado et al. [90]	*n* = 43 NASH patients *n* = 33 healthy controls (27–68 years)	increased α-TOH plasma levels in NASH patients	-
	Interventional	Violet et al. [91]	*n* = 6 female NASH patients(33–53 years) *n* = 10 female healthy controls (19–35 years)	alteration of α-TOH kinetics in women with obesity-associated hepatosteatosis compared to healthy controls, decreased release of α-TOH from the liver, lower α-TOH plasma level	2 mg α-TOH once
	Interventional	Sanyal et al. [92]	*n* = 167 NASH patients (*n* = 83 placebo, *n* = 84 α-TOH, 34–59 years)	α-TOH supplementation improves ALT, AST, lobular inflammation and NASH compared to placebo treated group	537 mg (800 IU) α-TOH per day or placebo for 2 years
	Interventional	Lavine et al. [93]	*n* = 11 NASH patients (<16 years)	α-TOH supplementation decreased ALT, AST, ALP	268–805 mg (400–1200 IU) α-TOH for 4–10 months
**MetS**					
	Observational	Ford et al. [94]	MetS patients and healthy controls from NHANES III (≥20 years)	lower α-TOH plasma level in MetS patients	-
	Observational	Beydoun et al. [95]	*n* = 3008–9099 participants from NHANES 2001–2006 (20–85 years)	higher α-TOH plasma level in MetS patients	-
	Observational	Yen et al. [96]	*n* = 72 MetS patients *n* = 105 healthy controls	α-TOH/TG plasma level remained unchanged	-
	Interventional	Mah et al. [97]	*n* = 10 MetS patients *n* = 10 healthy controls	MetS patients have lower α-TOH /lipid plasma level and lower α-TOH absorption and impaired hepatic trafficking compared to healthy subjects	15 mg α-TOH once

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
