# Peer review of "Cardiovascular and Metabolic Protection by Vitamin E: A Matter of Treatment Strategy?"

_antioxidants, 2020, doi:10.3390/antiox9100935_

Round 1

Reviewer 1 Report

Over the last decades, the extensive work of Peter Libby, Paul Ridker, Göran Hansson and others have contributed greatly to our understanding of the contribution of inflammation to the development of atherosclerosis and with the CANTOS study the success of an anti-inflammatory therapy was documented for the first time in a large cardiovascular endpoint study.

Vitamin E is known for its antioxidant and anti-inflammatory properties, and observational studies have repeatedly found a link between vitamin E deficiency and an increased risk of cardiovascular disease. However, it has not yet been convincingly demonstrated that supplementation with vitamin E can reduce the cardiovascular risk.

In this well written review, Melanie Ziegler and the co-authors summarize the relationships between vitamin E levels and various risk factors for cardiovascular diseases. In addition, they try to find explanations for why the connection of low vitamin E levels and cardiovascular risk found in observational studies can´t be used as successful treatment option in interventional studies. In this context they put forward the thesis that vitamin E application in acute therapy of myocardial infarction is likely to be more successful than in chronic supplementation.

Minor points:

  • Fig. 1 – due to the overlapping structures a-TOH is missing a CH3; perhaps the authors could orientate the TOHs in a similar way like the T3 molecules
  • Fig. 2 is a wonderful summary of the study results. It would be nice if the figure could be supported by a table with the most important studies, showing the case numbers, endpoints and for interventional studies the vitamin E dosage.
  • “based on clear evidence of LDL-C being a strong causal risk factor for CVD [71,72]” – Is 72 really a good reference for this statement, it´s only a commentary to ref. 81? I would prefer a paper like the EAS consensus statement: European Heart Journal, Volume 41, Issue 24, 21 June 2020, Pages 2313–2330
  • Is age itself really a risk factor for cardiovascular disease or is it only a surrogate parameter for the lifelong sum of the other risk factors?
  • Fatty Liver Disease: In a very recent publication (JCI Insight. 2020;5(1):e133309. https://doi.org/10.1172/jci.insight.133309.) Pierre-Christian Violet and colleagues showed very nicely that in women with hepatosteatosis vitamin E is trapped within the liver fat resulting in decreased vitamin E availability.
  • Although the authors cite publications that show common roots between AD and CAD, in my opinion this chapter doesn´t fit into this review.
  • Fig. 4 – perhaps it is a problem with my pdf reader but in this figure some letters are missing or not complete
  • Please actualize ref. 244: N Engl J Med 2019; 380(8):752-762

Reviewer 2 Report

Overall, this is a very well written and timely review article which should be of great interest to the readership of Antioxidants.

The article is clearly structured and easy to read.

The following GENERAL points should be addressed in the revised version of the paper:

  1. I would like to suggest that the authors shorten the introduction section a little bit. There are already many reviews articles availble in the literature regarding the chemistry, antioxidant and gene-regulatory activity of vitamin E.
  2. However, the conclusion section could be extended. The authors should come up with more detailed ideas how future studies on vitamin E in the context of chronic disease prevention should be better designed (in order to avoid a negative outcome).
  3. Chapter 7 entitled "Limitations" should be extended referring to both "MERITS and Limitations" of this article.

SPECIFIC points:

4. Potential vitamin E/vitamin K interactions should be described in more detail

5. Beyond the Hp2-2 genotype other genotypes (e.g. ApoE, PON-1 etc.; highly relevant to AD and CVD) may affect the outcome of vitamin E supplementation which should be discussed in more detail.

6. Furthermore the authors should consider in this context that proteins involved in specific vitamin E binding are not highly polymorphic and thus may not substantially influence inter-individual variation in response to vitamin E supplementation (doi: 10.1080/15216540400020346). Contrary, proteins involved in drug/lipid metabolism which indirectly influence vitamin E status are highly polymorphic, are likely to influence inter-individual variation and so are good candidates for vitamin E GWAS  studies. 
